# Dissipative Structures, Organisms and Evolution

**DOI:** 10.3390/e22111305

**Published:** 2020-11-16

**Authors:** Dilip K Kondepudi, Benjamin De Bari, James A. Dixon

**Affiliations:** 1Department of Chemistry, Wake Forest University, Winston-Salem, NC 27109, USA; 2Center for Ecological Study of Perception and Action, University of Connecticut, Storrs, CT 06269, USA; benjamin.de_bari@uconn.edu (B.D.B.); james.dixon@uconn.edu (J.A.D.); 3Department of Psychological Sciences, University of Connecticut, Storrs, CT 06269, USA

**Keywords:** nonequilibrium thermodynamics, dissipative structures, organism, entropy production, evolution

## Abstract

Self-organization in nonequilibrium systems has been known for over 50 years. Under nonequilibrium conditions, the state of a system can become unstable and a transition to an organized structure can occur. Such structures include oscillating chemical reactions and spatiotemporal patterns in chemical and other systems. Because entropy and free-energy dissipating irreversible processes generate and maintain these structures, these have been called dissipative structures. Our recent research revealed that some of these structures exhibit organism-like behavior, reinforcing the earlier expectation that the study of dissipative structures will provide insights into the nature of organisms and their origin. In this article, we summarize our study of organism-like behavior in electrically and chemically driven systems. The highly complex behavior of these systems shows the time evolution to states of higher entropy production. Using these systems as an example, we present some concepts that give us an understanding of biological organisms and their evolution.

## 1. Introduction

### 1.1. Thermodynamics

The field of thermodynamics was born out of Sadi Carnot’s fascination with the recently developed steam engine [1] that generated mechanical motion using heat. Central to understanding these machines was the nature of heat. In France, at the time of Carnot, heat was thought of as a fluid called Caloric. In his only publication, Reflections on the Motive Power of Fire [1], Carnot expounded on the fundamental limitations of heat engines and, in doing so, he discovered fundamental nature of irreversibility in natural processes. The central result of Carnot’s work was that a reversible heat engine had the highest efficiency and this efficiency depended only on the temperatures between which the heat engine operated, regardless of the particular mechanism of the heat engine. Clausius [2] expanded Carnot’s work into the concept of entropy, S. In this classical theory of entropy, entropy was a function of state, expressed in terms of state veriables, but its relation to irreversible processes that generated is was not explicit.

The modern formulation of thermodynamics was extended to include irreversible processes that drive changes in the system [3,4,5,6] through the concepts of thermodynamic forces and flows. In this formulation, change in entropy in time dt is written as:(1)$$dS=\;\;d_{i}S+d_{e}S$$
where $d_{i}S$ is the entropy production due to irreversible processes, which according to the Second law is always positive, and $d_{e}S$ is due to the exchange of energy and matter with the system’s exterior. $d_{i}S$ is directly related to the thermodynamic processes occurring within the system, which are describable in terms of thermodynamic forces and flows. Thermodynamic forces are gradients of intensive variables *X* (e.g., pressure, temperature, and chemical affinity) divided by temperature, and flows are time-derivatives of the corresponding extensive variable *J* (volume, heat, and reaction rate, respectively). The irreversible entropy production per unit volume is calculated as the sum of the product of all forces and flows in the system:(2)$$\frac{d_{i}s}{dt}=\;\sigma =\;\underset{i}{\sum }X_{i}J_{i}$$
where s is the entropy density. By integrating *d_i_s/dt* over the volume of the system, the total entropy production *d_i_S/dt* is obtained. Flows are driven by forces, as in the flow of heat driven by a temperature gradient, though forces are not functions of the forces alone. Flows may depend on other variables, such as catalysts. Flow rates can vary due to system properties independent of the driving force, such as heat capacity in the case of thermal flows and catalysts in the case of chemical reactions. Critically, Equation (2) establishes that the entropy is a direct function of the processes driving changes in a thermodynamic system.

### 1.2. Extremum Principles in Thermodynamics and the Entropy Production

We note that thermodynamics stands in contrast to mechanics. In mechanics, if the initial conditions and forces are known, the time evolution of the system is entirely determined. Thermodynamics, on the other hand, only predicts the final state, not the path towards it and, in this sense, it is end-directed. In the sections below, we will describe how this end directedness gives rise to organism-like behavior in non-equilibrium systems.

In classical thermodynamics, extremum principles take different forms, depending on the system. Isolated system evolves to a state of maximum entropy; for closed systems that are maintained at constant temperature, T, and volume, *V*, the Helmhotz energy evolves to its minimum. In closed systems at constant pressure, p, and T, Gibbs energy evolves to its minimum [6]. In each case, the theory only predicts the final state, not the dynamic path to it. We can guarantee this end-state, but classical thermodynamics does not predict the exact nature of the processes that will take it there or how long it will take. Classical thermodynamics, being a theory of states, not processes, does not have time evolution of any variable in its formulation (for which reason, it might even be appropriate to call it thermostatics!)

Modern thermodynamics is a theory of processes and it contains the time evolution of entropy, i.e., the rate of entropy production, and time evolution of flows and forces. In modern thermodynamics too there are extremum principles, but these are for the rate of entropy production (which is often referred to simply as entropy production) [4,5,6]. One such principle is Prigogine’s minimal entropy production principle [4,5,6], which states that for a system close to equilibrium, where flows are linear functions of forces—the so-called linear regime—the steady-state will be that which minimizes the rate of entropy production. If the system is not allowed to reach the state of equilibrium at which the entropy production is zero, it evolves to its minimum possible value. Moreover, as is well known, in the linear regime we have the celebrated Onsager reciprocal relations [4,5,6].

A more recent hypothesis that has been of much interest, and some controversy, is the maximum entropy production principle (MEPP) that prescribes that a non-equilibrium system will tend towards processes and states for which the rate of entropy production is maximal [7,8,9,10,11,12]. MEPP is not applicable to all non-equilibrium systems, as one can give examples of situations in which it is not valid. For example, in a resistor subject to a constant voltage V, the total entropy production equals VI/T, in which I is the current and T is the temperature. When the voltage is turned on, consequent heating increases the resistivity resulting in a decrease in I and an increase in T. As a result, the entropy production decreases, contrary to MEPP which predicts an increase. On the other hand, MEPP has been demonstrated to be valid for many other non-equilibrium systems. Some have recently proposed conditions necessary for MEPP to apply [10,11,12]. Extremum principles can be powerful tools for predicting the time evolution of a system to its “end state” independent of knowledge of the complex processes at work within the system. Non-equilibrium thermodynamics has the promise of being both a description of processes within a system, in terms of forces and flows, and the end-states of systems, in terms state variables and the rate of entropy production.

### 1.3. Dissipative Structures

The most important advance in non-equilibrium thermodynamics is the study of self-organization. This is a remarkable phenomenon which shows how irreversible processes can lead to spontaneous organization and creation of ordered states: some far-from-equilibrium systems with nonlinear relations between forces and flows can develop processes that are structured in space (e.g., fluid convection rolls, chemical patterns) or time (e.g., oscillating chemical reactions) that persist through dissipative processes [13,14,15]. The development of structures in these systems tends to coincide with an increased rate of entropy production required for the maintenance of such structures [13,14,16]. Since these organized state or structures are maintained through dissipation of free energy and generation of entropy, they have come to be known as dissipative structures. As will be discussed in length below, dissipative structures are in stark contrast to machines or designed structures whose structure is by design originating form an external source, not spontaneous and from within the system as is the case with dissipative structures. Some of the main characteristics of dissipative structures are as follows.

#### 1.3.1. Amplification of Fluctuations and Establishment of a New Structure

Under far-from-equilibrium conditions, a state can become unstable. When this happens, the system can make a transition to an organized state, a dissipative structure. In a thermodynamic system all variables undergo thermal fluctuation. When a system is in a stable state these fluctuations are damped by irreversible processes. When the system is crosses a point of instability, a particular type of fluctuation is not damped, and it will grow. The growth of a fluctuation drives the system to a new state [6,17,18,19]. This growth is driven by autocatalytic processes, wherein a product of a process (e.g., light, charges, or chemical compounds) catalyzes its own production. Though the mechanism of growth of a fluctuation is within the system, which fluctuation will grow and what new structure will arise depends on the boundary conditions i.e., the environment. 

#### 1.3.2. Spontaneous Symmetry Breaking

More often than not dissipative structures arise out of a symmetry-breaking transition [17,18,19,20,21,22]. In such a transition, the state to which the system evolves does not have the symmetry of the processes that generate it. In mathematical terms, it means the solution to a differential equation does not have the symmetry of the equations. Due to the symmetry of the equations, symmetry breaking leads to a multiplicity of solutions, each solution related to another through a symmetry operation. In such situations, at the point of transition to a dissipative structure, the system has several states to which it can make a transition. Which one it will transition to will depend on environmental or boundary conditions or some small asymmetries influencing the time evolution of the system. It could be said that in symmetry breaking transitions, the environment imprints on the structure. Such phenomena are compared, in Section 4.3, to the epigenetic pathways related environmental factors to genetic fluctuations.

#### 1.3.3. Selection of States and Sensitivity

The selection of states through environmental factors make the system very sensitive to its environment, known as cooperative sensitivity [22,23,24]. Through such a mechanism, the internal structure of a dissipative structure becomes correlated with external factors. In our earlier work, a theory of sensitivity in symmetry breaking transitions was developed [22,23,24]. In the case of a two-fold symmetry, such as chiral symmetry, it was shown that the system can become extraordinarily sensitive to very small chiral influences during a slow passage through the transition point [6,20,21,22]. It was also noted that, through such a mechanism, internal states of small living cells can become correlated with the direction of gravitational field, thus providing a detection mechanism for gravity.

#### 1.3.4. Self-Healing

Dissipative structures are stable to perturbations. If a spatial or a temporal structure is perturbed, in due course the structure is reestablished [14,25,26,27]. This implies “self-healing”. When there is damage to the structure, the structure is restored. Since the irreversible processes that created a dissipative structure are within the structure, the system has the ability to restore the structure and “heal” damages. This aspect is an important characteristic of biological organisms.

### 1.4. Dissipative Structures and Life

The structure and processes of living organisms is driven and maintained by irreversible processes. Though the particular process of self-organization that has resulted in the familiar forms of life is not known, living organisms are undoubtedly dissipative structures, given that their existence and behaviors are sustained by continuous flows of energy and matter [14,17,28,29]. Modern thermodynamics has the tools of forces, flows, and rates of entropy production that can be used to understand biological processes.

In line with a number of theorists [14,17,28,29], we hold that living systems are a subset of the class of dissipative structures, and that their origin, behaviors, and evolution are critically related to their thermodynamic properties. Recently, we have demonstrated life-like behaviors in an electrical dissipative structure [25,26,27,30,31,32] and a chemical dissipative structure [33,34]. These Bio-analogue dissipative structures (BDS) provide a compelling scientific context in which physics and the life sciences inform one another to develop explanations. Herein, we will propose that such BDS provide insight into the origins and evolution of living systems and prescribe a thermodynamic framework for understanding the origin of the behavior we see in living organisms.

## 2. Machines, Dissipative Structures, and Organisms

The theory of organism we present here is founded in the realization that mechanics is the science of machines, while in thermodynamics lies the physics of organisms. It means thermodynamics is the foundational science of organisms. To understand this let us look at the relationship between mechanics and machines.

Humans have been inventing machines since the dawn of civilization. As machines became more complex and became substitutes for animal and human labor, the notion (attributed to Réne Decartes [35]) that animals and organisms are simply complex machines has become integral to the way we think about organisms. Such thinking has deep roots in the rise of Newtonian mechanics, and later version, Lagrangian and Hamiltonian mechanics. Their success in explaining motion in general, and planetary motion using the law of gravitation in particular, ushered a new era in understanding natural phenomena. The laws of mechanics are time reversible: all motion and its time-reversed version are in accord with the laws of mechanics. However, it must be noted that from the outset, mechanics did not give us an explanation or any insight into biological phenomena. It had no means to distinguish living and dead matter. The great success of both classical and quantum mechanics lies in their role as the foundation for the invention of machines of increasing complexity, machines that have transformed human life in unimagined ways, machines of immense power, machines that vastly outperform humans and animals in performing specific tasks. Even machines that were inspired by biological organisms, such as airplanes and submarines, are entirely different form the organisms that inspired them: though it flies, an airplane is nothing like a bird. The difference is fundamental, in the very processes that generate the two distinct systems. In contrast, dissipative structures have much in common with organisms. We now recognize that organisms are highly complex dissipative structures, not highly complex machines. Let us look at characteristic features of machines on the one hand and dissipative structures and organisms on the other to see that organisms are indeed more complex versions of non-living dissipative structures.

Central to the foundational difference between organisms and machines is the role of irreversible processes. Machines ideally have minimal dissipative or irreversible processes, while dissipative structures and organisms run on intrinsically irreversible processes. Entropy production in a machine is a measure of inefficiency, where energy is converted to lost heat rather than mechanical work. In dissipative structures, entropy producing processes maintain structure and function, as in the irreversible chemical transformations that drive spatiotemporal structuring in a chemical clock. Similarly, the endogenous chemical cycles of organisms (metabolic, behavioral) are driven by irreversible entropy-producing processes.

Whereas machines are created by design, prescribing the composition and relations of all parts, organisms and dissipative structures emerge spontaneously through complex dissipative processes without externally prescribed instructions. The structure of parts, their relations, and even the distinction between parts are influenced by the contexts in which the organism develops.

Machines, unlike organisms, are what Rosen [36] termed fractionable: the structure and function of component parts are isomorphic such that a machine can be neatly decomposed into its constituents and each constituent performs a specific function. Organisms are not fractionable in this sense; the boundaries between parts are ambiguous (e.g., the integration of the nervous and vascular system), and the functional role of a given constituent is flexible for changing contexts within and external to the organism (e.g., the function of a muscle in different postures or relative to different activities). Mechanics is the physics of machines—the physics of fractionable systems—and is thus inadequate for a complete explanation of biological non-fractionable systems.

Rosen’s fractionability [36] has important consequences: for machines, an analysis of their parts and associated functions also gives us the knowledge of how to make that machines. Such is not the case for organisms. Despite a detailed understanding of a given biological process, we do not presently have the means for taking the constituent parts and assembling them in a way that is functionally equivalent to its natural counterpart. This is because the way we’ve attempted to understand biological processes is to break them into fractioned component parts with isomorphic structure-function relationships. As noted, however, biology is critically non-fractionable, and so analyzing an organism it as if it were a machine of fractionable parts does not enable synthesis of an organism.

The origin of function in machines and organisms are in stark contrast. A machine’s function is derived externally, either from an engineer’s prescription or in how it is used, whether as its intended use or not. Function within dissipative structures is intrinsic, identifiable in their end-directed time evolution. Some dissipative structures (expanded on below) exhibit morphological and behavioral changes that drive the system to states of greater entropy production [25,26,27,30,31,32]. In our electrical dissipative structure, this manifests in an ‘energy-seeking’ behavior where tree-like structures of conducting spheres move through their environment, directed by electrical gradients, to regions of higher concentration of charges. This energy-seeking behavior can be paralleled with that of living organisms which similarly navigate their environments using embedding energetic fields (light, sound, chemical) to find energetic resources that sustain their structure.

The functionality of biology is, we assert, more like the intrinsic end-directed time evolution of dissipative structures than the externally prescribed functionality of machines. When a complex dissipative structure is formed, it is formed as a whole with spatiotemporal correlations and structure; the process is not a serial aggregation of parts. However, in understanding the behavior of a dissipative structure, one might attribute functionnality just as we do with organisms. For example, in the tree-structure we have studied, before the formation of a tree, the disconnected random beads are similar in their behavior; the location of each bead has no particular significance. However, once the tree is formed, the connectivity (the number of beads a bead is in contact with) is different. A bead is generally in contact with, one, two or three other beads. The beads at the tip of the tree are the collectors of charge that is being sprayed from the electrode tip. Our experiments clearly indicated that the charge flows from the source electrode to the tips of the trees, not to all the beads in the tree [25]. Now we may classify the beads in a tree into three types, the “collector” beads (connected to only one other bead), transmission beads (beads in contact with two other beads) and branching beads (beads in contact with three other beads), and assign a functionality, such as the collection of charge, transmission of charge, and branch transmission of charge, to each type. Other types of functionality are attributed to biochemical [37,38] dissipative structures. Such functionality is not externally imposed but intrinsic to the structure, which in turn is an outcome of the system’s intrinsic end-directed evolution. 

These and many other distinctions between organisms and machines point to thermodynamics as the foundational science for both dissipative structures and organisms. It must be noted that, while machines and organisms are fundamentally different classes of systems, machines are richly integrated into human life. Machines may be understood as parts of a broader non-equilibrium self-organizing system, but absent that embedding they are fundamentally different. It is interesting to note, however, that industry has advanced humanity’s entropy production, sometimes autocatalytically. For example, oil extracted from the ground is used to power combustion engines to extract more oil and produce more combustion-powered machinery. Table 1 summarizes important features that differentiate machines on the one hand and dissipative structures and organisms on the other. The tools for understanding dissipative structures then will be informative for explaining biological phenomena. In the context of evolution, we identify that the core properties of dissipative structures in Section 1.3.1, Section 1.3.2, Section 1.3.3 and Section 1.3.4 will be relevant to establishing the relationship between thermodynamics and biological evolution.

## 3. Bio-Analogue Dissipative Structures: The Missing Link

Given that organisms are dissipative structures, there is likely a continuum of systems that spans from non-living to living dissipative systems. Dissipative structures studied for many decades starting from the 1970s provided insight into how biological form and pattern formation might emerge from nonequilibrium chemical systems (with appropriate autocatalytic and other mechanisms) and how irreversible chemical processes might provide “chemical clocks” and give rise to periodic phenomena in living systems. However, they did not have any bio-analogues that behaved like living systems and exhibited end-directed behavior such as “foraging” and moving towards the source of energy that sustains it. In our recent work, we have identified such a system which is electrically driven. It is a “missing link” between dissipative structures and organisms: it exhibits organisms-like behavior, but it is not a living organism. Such non-living dissipative structures, we propose, can serve as examples of the class of systems that bridges non-living and living dissipative systems. In these complex systems, several thermodynamic forces and flow are coupled. In our system the structure’s motion is a consequence of current flow and the convection currents in the oil.

Our studies of such non-living dissipative structures have demonstrated a host of biologically plausible behaviors, which we discuss here. These bio-analogue dissipative structures (BDS) serve as minimal models for biology and lend a framework for understanding the physical foundations for a physics of organism and evolution of life. Dissipative structures have elsewhere been theoretically proposed as models of psychological systems, including the study of human motor control [39], especially in the production of rhythmic movements [40].

We have studied two dissipative structures, an electrical and a chemical system, which we classify as self-organized foraging implementations (SOFIs). Our electrical system (E-SOFI) produces an array of life-like properties, including foraging [25,26,27,31]), end-directed behavior [25,26,27,31], structure maintenance and self-healing [25,26,27], and even functional coordination [30,32,41]. The chemical system (C-SOFI) demonstrates flocking behavior [33,34] sensitivity to external thermal and magnetic fields [33], and collective navigation of environments [34]. Several other groups in the field of dynamic self-assembly (DySA) have demonstrated similar life-like behaviors, including chemotaxis [42], collective dynamics [42,43,44,45], and self-healing [46].

### 3.1. The E-SOFI

The E-SOFI (Figure 1) consists of metal beads in shallow oil subject to a high electrical voltage across a suspended source electrode and a grounding ring in the dish. The electrical forces drive the formation of strings of beads, called “trees”, that maintain contact with the grounding electrode. The trees are rudimentarily end-directed to increase the rate of entropy production [25,26,27]. The system self-selects for morphologies and behaviors that maximize the current through the system. In this system, the rate of total entropy production (REP) (Equation (3)):(3)$$\frac{d_{i}S}{dt}\;=\;\frac{V*I\left(t\right)}{T}$$
where *V* is the voltage, *I(t)* is the total current at time *t*, and *T* is the temperature of the system, which was found to be uniform and fairly constant within a few percent. This intrinsic end-directedness manifests in the trees’ foraging activities, and in the repair of trees subject to perturbation. Trees move through the dish to collect charges that build up on the surface of the oil [31] while the current flowing through them creates the forces that maintain their structure. The trees thus forage, much like organisms, by moving through their environment to collect the energetic resources that keep them intact. When trees are perturbed, for example by breaking them up with an insulating rod, the system will reform into new trees. While the morphology may differ post-perturbation, the new system produces approximately the same REP [27].

Trees exhibit functional interdependence as well, coordinating their activities to increase the REP [30,32,41]. Trees that are perturbed from steady-state dynamics, resulting in a decrease in the current, will relax back to a new steady-state that increases the REP to approximately the same pre-perturbed level. Critically, during the relaxation epoch, their activities are coordinated, suggesting that the trees co-regulate their motion to produce a behavioral state of increased REP. Trees constrained to behave as oscillators exhibit relative-phase coordinative modes characteristic of biological coupled oscillators [41], and can exhibit compensatory behaviors for a coupled partner that becomes constrained [32].

The observed tree formation and the associated increase in total current could also be seen form the viewpoint of Adrian Bejan’s constructal law, which states that:

“For a finite-size system to persist in time (to live), it must evolve in such a way that it provides easier access to the imposed currents that flow through it” [47,48,49].

One could say that the behavior of E-SOFI is in accordance with the constructal law. In this one instance, whether we choose a constructal law or the rate of entropy production seems a matter of choice. Our choice of entropy production is because constructal law applies to the flow of current but has no direct relationship to a fundamental and universal thermodynamic quantity such as entropy. Constructal law has been discussed with much generality in which “flow” refers wide range of physical as well as non-physical entities such as knowledge [49].

### 3.2. The C-SOFI

The C-SOFI consists of irregularly shaped benzoquinone (BQ) pellets floating at the air-water interface in a petri dish. BQ dissolves into the water, changing the surface tension around the pellet, propelling it across the surface [33,34]. When multiple pellets interact, they tend to aggregate into a cluster of pellets that move together through the dish, called a “flock” (Figure 2). This flock tends to move across the surface as a singular entity, though it may fluctuate in size and number of pellets. The flock tends to move to regions of the dish with lower aqueous BQ concentration, which leads to faster dissolution and thus greater REP (Equation (4)) [33]. In this system, the total entropy production is
(4)$$\frac{d_{i}S}{dt}=\;\;AVk\left(C_{0}-\;C\left(t\right)\right)$$
where *A* is chemical affinity, *V* is volume, *k* is the dissolution rate, $C_{0}$ is the saturation concentration of BQ, and *C(t)* is the aqueous BQ concentration at time *t*. The C-SOFI forages for circumstances that increase the REP, and thus abides by directly analogous thermodynamic processes as the E-SOFI. The C-SOFI can be similarly couched as end-directed towards states of greater REP.

The flocks demonstrate collective sensitivity to exogenous thermal and magnetic fields. Flocks will swim towards or away from hot and cold probes respectively, moving as a single entity [33]. High temperature likely increases the dissolution rate and thus the REP, making the thermotaxis functional. When a single pellet is embedded with ferrous material, the flock can become collectively sensitive to a weak magnetic field. A weak magnetic force can be created by positioning a magnet at a height above the dish that only slightly affects the motion of a single BQ swimmer (i.e., the swimmer may have a slight bias towards the magnet, but swims freely through most of the dish). When multiple pellets are then added and a flock forms, the entire flock moves toward and remains under the magnet [33]. This sensitivity to weak exogenous fields may be compared to the perceptual sensitivity of organisms. For example, sighted organisms are sensitive to weak electromagnetic fields by virtue of their self-organized structure [35]. Light-driven chemical swimmers have been developed that similarly exhibit complex collective dynamics [44].

### 3.3. Fractionability in Dissipative Structures

A distinction was drawn between fractionable mechanical systems and non-fractionable biological ones. If dissipative structures are to be a suitable model of living systems, they ought to be similarly non-fractionable. The E- and C-SOFI do have properties that suggest they are essentially non-fractionable much like living systems. In the E-SOFI, the self-healing phenomenon [25,27] highlights the clear structure–function ambiguity characteristic of non-fractionable living systems. Recall that trees, after being subjected to destabilizing perturbations, will reform into new trees of potentially different morphologies that re-stabilize the REP. The functionality of the structure is not tied to the configuration of component parts, but rather is in achieving a certain relation to the embedding field, here minimizing resistance to allow flow.

The functionality of an E-SOFI tree is not isomorphic with its structure; there is no one specific tree to achieve a function. This is apparent in the results that trees can form into a variety of shapes in different trials but approach the same current for the same applied voltage (Figure 3). The function of foraging for charges and sustaining the structure is achieved in many different ways through self-organization. The variability of structure that achieves a required function is a characteristic of living organisms. Higher variability gives the organism better survivability.

The non-fractionability manifests also in the way the E-SOFI can be built. Rosen identified that for fractionable systems understanding a process entails being able to synthesize it. We can easily arrange beads into the same configuration that it might arrive at naturally, does this suggest the E-SOFI is fractionable? While we can replicate the structure of a given tree, the functionality, evaluated in terms of its conductivity, of that tree depends on its history and present context. The morphology of a tree has contributions from electrical forces, and complex fluid dynamics, both of which likely interact. The fluid dynamics in particular are very likely to change from trial to trial, and such contextual changes may result in differently functional morphologies. Figure 4 presents examples of different morphologies developed over different trials. Dividing a self-organized structure into component parts and trying to assemble these component parts to obtain a structure may or may not result in a system remaining in the assembled structure. For example, if the beads are assembled into a particular tree when the voltage is turned off, that tree may or may not maintain its structure when the voltage is turned on. As soon as the voltage is turned on, convection currents begin, and these may move the beads into a new configuration. When the beads self-organize, the final shape of the tree is a result of a very complex time evolution of the convection currents and the forces that move the beads [25,27]. Fractionable machines, on the other hand, are not likely to spontaneously change in a way that maintains or improves function after assembly. Most spontaneous changes in the structure of machines are characterized as breakage.

The C-SOFI perhaps more dramatically demonstrates the non-fractionability of these systems as the configuration of the flock is continually changing as are its constituents, with pellets entering and leaving the flock throughout time. Nevertheless, functional characteristics of the flock such as taxis are maintained at the level of the flock. The sensitivity to magnetic fields similarly demonstrates the structure-function ambiguity. Under a traditional analysis, we might distinguish the functional roles of the ferrous and non-ferrous pellets, the former being a “sensor” and the latter being structural components. Interestingly, the ferrous pellet only appears to function as a sensor when it is embedded in the flock. Its functionality emerges under certain contexts rather than being intrinsic to it. Insofar as living systems are critically non-fractionable, scientific attempts to understand biological functioning will be greatly informed by the study of similarly non-fractionable dissipative structures.

## 4. Dissipative Structures and Evolution

### 4.1. Evolution

Evolution is broad as both a phenomenon and concept, and these two aspects are not always in concert. The “modern synthesis”, combining Mendelian genetics with Darwinian natural selection, characterized evolution as the inter-generational change in frequencies of alleles within a population [50,51]. Such theories focused on the microscopic genetic factors (though the nature of DNA was unknown early on) as the objects of evolution. More contemporary definitions expanded to encompass the phenotypes of developed organisms:

“Evolution may be defined as any net directional change or any cumulative change in the characteristics of organisms or populations over many generations—in other words, descent with modification… It explicitly includes the origin as well as the spread of alleles, variants, trait values, or character states” [52].

Conservatively, evolution consists of at least the inter-generational change of organisms and the forces that drive such changes. While lacking an all-encompassing definition of evolution, we focus on properties likely to be included in any adequate definition: (1) micro-scale (i.e., genetic) mutations with macro-scale (i.e., phenotypic) consequences, (2) adaptive selection of traits by the environment, and (3) self-replication through reproduction with inheritance of traits. Together these processes drive the production of organisms whose traits may vary across generations as a joint function of stochastic mutations and selective environmental pressure. Processes (1) and (2) in biological evolution are mirrored in dissipative structures through phenomena of (1) fluctuation-amplification and (2) non-equilibrium cooperative sensitivity. Process (3) has, to our knowledge, few present instantiations in BDS (e.g., [43]), though we speculate on the properties that may enable such a process.

### 4.2. The Continuum of Dissipative Structures

The BDS described herein are offered as examples of systems that may bridge living and non-living dissipative structures. These BDS may be considered a form of para-life; analagous to but distinct from living systems. In the origin and evolution of life, similar types of systems may have been the progenitors of organisms. Given the shared physical foundations of organisms and BDS, the tools of non-equilibrium thermodynamics likely can be used to effectively understand biological evolution. Biological evolution has frequently been couched in thermodynamic terms (for a brief historical review, see [28]), with much focus on information entropy [29,53,54], or on the hypothesis that a MEPP is a driving force of evolution (e.g., [7,9,18,29,55,56]). Such a theory is not without dispute (e.g., [57]), and our own empirical work offers little to support or oppose such far-reaching hypotheses, so we remain largely agnostic on the subject. Nevertheless, we offer that some tools from non-equilibrium thermodynamics can inform our study of evolution. We will discuss how the theory of dissipative structures, especially fluctuation amplification and autocatalysis, can be used in the context of evolution. We will present some evolution-like phenomena that can be observed in the E-SOFI that support the notion that dissipative structures instantiate fundamental characteristics of biology.

### 4.3. Fluctuation-Amplification and Genetic Mutation

Contemporary understanding of biological evolution identifies the importance of microscopic fluctuations in the structure of DNA to produce variability in phenotypes that can be selected for, leading to evolution of subsequent generations [50,51,52]. These fluctuations, as novel genetic sequences, are amplified by selective processes that result in (among many other consequences) the production of more of the same genetic sequences. The selective processes at the macro-scale, constituted in the interaction between organism, behavior, and environmental properties, lead to the propagation of genetic material through reproduction. The phenotype of an organism stems in part from micro-scale genetic factors and their fluctuations that change the DNA. Similar characteristics can be observed in non-living dissipative structures, whose properties depend on micro-scale fluctuations.

Dissipative structures are defined by the emergence of new dynamical modes in the structure or activity of a system as a non-equilibrium control parameter is varied [6,13,14,15,16]. These processes can be described by dynamical equations of the macroscopic forces and flows in the system, with bifurcations in the solution space corresponding to the emergence of dynamical modes. Dissipative structures are typically characterized by nonlinear equations, which give rise to a multiplicity of stable modes. In the range near the critical value of the bifurcation parameter, systems are particularly sensitive to microscopic fluctuations that can drive the system to one of the stable modes or another.

Whereas, in near-equilibrium systems, fluctuations are typically counteracted by relaxation processes (e.g., the averaging of particle velocities in an isolated system of gas). The non-linear interactions and far-from-equilibrium nature of dissipative structures can lead to the amplification of fluctuations and selection of one of many possible states, as noted above in Section 1.3.1. As a specific example, consider convective rolls in a fluid subject to a thermal gradient. If we observe a single convective cell, the fluid may flow clockwise or counterclockwise, both of which are equally stable dynamical modes [14,15,58]. The direction of flow is determined by microscopic velocity fluctuations occurring in the fluid at the time of formation of the convective cells. Small imperfections in the conducting plates that favor one fluid-flow direction, clockwise or counterclockwise, could determine direction of fluid flow. The intrinsic instability around bifurcation points means that the development of a dissipative structure is subject to probabilistic influences, making the history of the system and the boundary effects determinants the present state [14,17,18]. Not unlike biology then, microscopic fluctuations can drive differentiation of macroscopic phenotype.

Chemical systems similarly can be shaped by microscopic fluctuations. When a complex oscillating reaction like the Belousov–Zhabotinsky reaction is prepared in a dish, a wide variety of spatial patterns cam emerge, often termed “Turing patterns”, which can vary with differences in initial conditions [13,14,15]. These structures are a result of coupling between chemical reactions and diffusion [59,60]. Computer simulations of an autocatalytic reaction happening within two volumes that exchange compounds reveal multiple steady-state solutions with asymmetrical concentrations of the compounds [18]. These steady-states are equally probable, and the emergence of one over the other is driven by microscopic fluctuations near the critical point.

Prigogine summarizes this interplay of microscopic fluctuations and differentiation of phenotype as “order through fluctuations” [17,18]. Within nonequilibrium systems, there is an intrinsic interaction between the structure, given by the dynamical equations and relation between parameters, the function, given by the various dynamical modes (steady states, limit cycles), and the fluctuations that trigger instabilities (Figure 5). Biological evolution may be characterized by a similar system where function may range from the processes within an organism to its behavior, structure is the composition of the organism, and fluctuations may be due to genetic variation or epigenetic environmental factors. A similar structure of causality in developmental biology was advocated for by Gilbert Gottlieb [61] known as probabilistic epigenesis. Throughout the many scales of biological development (e.g., genetic, cellular, intercellular, neural, behavioral, and environmental) fluctuations at one scale have bidirectional effects across scales, potentially driving cascading consequences on structure and function throughout the system.

### 4.4. Cooperative Sensitivity—Environmental Selection

The intrinsic instability near critical points in dissipative structures can render them sensitive to factors originating outside the system. As outlined above, endogenous fluctuations around the critical point drive the differentiation of dynamical modes in dissipative structures. Some external events, processes, or boundary conditions can similarly introduce fluctuations that bias the selection of one mode in a multi-stable state. The amplification of fluctuations in dissipative structures results for different reasons depending on the system, but there is generally an autocatalytic process at the core of amplifying fluctuations. Whereas thermal fluctuations in an equilibrium system would likely damp out the influence of an external field, this cooperativity can amplify its effect. Dissipative structures then have an intrinsic sensitivity to the contexts in which they develop and exist.

This context-sensitivity mirrors the adaptive evolution of organisms, bearing characteristics reflective of the habitat they co-evolved with. Genes may also be activated in an organism by virtue of myriad physiological and environmental factors through epigenetic pathways [61,62]. For example, *Drosophila Melanogaster* reared at different temperatures exhibited developmental and inter-generational morphological differences [63]. This suggestion is in contrast to the blind selection much of evolutionary theory assumes, where changes in genotype (and consequently phenotype) are merely stochastic and are selected for solely by the reproductive success of the organism [64]. While the boundary conditions and processes governing selective adaptation in biology are much more complex than in current BDS, the fundamental process of external factors introducing or biasing fluctuations that change the phenotype of the system is shared in both living and non-living self-organizing systems.

### 4.5. Autocatalysis

A process closely related to fluctuation-amplification is autocatalysis, or self-production. Autocatalysis manifests in nearly all self-organizing phenomena and a variety of thermodynamic systems, including lasers, electrical dissipative structures, and oscillating chemical reactions. In each case, a thermodynamic process, e.g., light emission, electrical conduction, or chemical reaction, is driven by the products of that process. In a laser, photons with a given wavelength stimulate the emission of other photons at the same wavelength, leading to the multiplicative production of coherent light-waves. In the E-SOFI discussed above, the conduction of charges through the beads to ground creates the forces that maintain a polarity imbalance between the trees and the surrounding charge distribution, driving the flow to the beads. In a chemical process, a given compound can catalyze its own production through a reaction with another compound. In each example, autocatalysis drives self-organization in the system, i.e., laser light, tree structures, and oscillating chemical reactions respectively.

Autocatalysis is apparent in living systems as well, and is core to our understanding of biological evolution and inheritance. Autocatalysis is most clearly present in the processes associated with genetic transcription and replication (e.g., [65]). DNA molecules participate in a complex reaction network that ultimately results in more of the same molecules being produced. Autocatalysis is also present at a more macro-scale: simply put, organisms beget more organisms of (roughly) the same type. There are sets of reproductive processes in organisms that “transform” an existing organism into one or more of the same organism. The fission of a bacterium is particularly illustrative, as a single cell undergoes a complex process of genetic replication, followed by dramatic morphological division, resulting in two bacteria with near-copies of the genetic material [66,67]. Coarse graining in thermodynamic terms, we see a complex nonequilibrium system (a cell) exhibits a critical bifurcation that drives the emergence of a new reaction network (onset of reproductive processes) that drives the autocatalysis of complex molecules (copying of DNA) and the production of new spatial structures (cell fission). An autocatalytic reaction network can be described as a cycle of reaction steps, not unlike the life-cycle of an organism: for each, a complex system cycles through a set of transformations that result in the repetition of certain structures [14]. While self-replication of the kind seen in organisms is not present in current BDS, it is possible that sufficiently complex artificial systems will be developed that capture such reproductive processes.

### 4.6. Developmental Context Sensitivity—Empirical Results

Thus far we have summarized some examples of the evolution-like properties of dissipative structures. Here, we expand with novel empirical results that highlight such biologically plausible features intrinsic to dissipative structures. As discussed above, fluctuation amplification and autocatalysis can lead to context sensitivity through external influence during critical bifurcations in dissipative structures, in a way that mirrors organisms’ adaptive compatibility with their environments. Due to the cooperativity and long-range correlations within non-equilibrium systems [21,22,23,24] weak external fields have increased interactive effects compared to equilibrium systems. Such external energy fields can bias the emergence of certain modes in multi-stable chemical dissipative structures [21,22,23,24]. In some systems it has been found that MEPP selects among multi-stable states [11,41]. Dissipative structures thus have an intrinsic context-dependence such that their development is rudimentarily adapted for the circumstances. We draw a direct analogy to the adaptive changes wrought by biological evolution.

We have demonstrated an analogue of this empirically in the E-SOFI, which can exhibit morphological variants in the presence of weak external magnetic fields. We prepared the E-SOFI with a set of 15 total beads, 10 aluminum, and five chrome. Chrome beads are sensitive to magnetic fields while aluminum beads are not. Beads were distributed approximately uniformly in the center of the dish prior to turning the system on (Figure 6). We ran the system for brief (i.e., less than 60 s) trials to observe tree formation, running until stable structures formed. Trials were performed in the absence or presence of a weak magnetic field, and we compared the morphology of trees indexed by the relative position of the chrome beads.

#### 4.6.1. Methods

A magnet was set up on a moveable arm below the dish. To ensure the magnetic field was sufficiently weak, the magnet was positioned at a distance below the dish such that lateral movements of the magnet could not move a single chrome bead while the system was off. The magnet forces were thus not strong enough to overcome frictional and inertial forces on the bead. The magnet was positioned approximately halfway between the grounding electrode and the source electrode (Figure 6). Ten trials were performed with the magnet removed from the dish, and ten trials were performed at the weak-field threshold. The system was run until stable trees formed (some beads did not connect to ground or incorporate into trees).

We quantified the morphology with the relative position of chrome beads in the trees by counting beads from the grounding electrode up to the chrome bead (e.g., if there were 5 beads between the grounding electrode and the chrome bead, the chrome bead was in position 6). Chrome beads that did not incorporate into trees are excluded from analysis. We expected that chrome beads should occupy greater position numbers (i.e., be further out on the tree) in the presence of the magnet than in the absence of the magnet.

#### 4.6.2. Results

An independent samples t-test revealed a significant difference (*t*(98.934) = 2.3967, *p* < 0.05) between the position of chrome beads in the presence (*M* = 6.627, *SD* = 3.053) and absence (*M =* 5.22, *SD* = 2.916) of the magnet. The chrome beads tended to be further out in the trees, and thus closer to the magnet. The morphology of the trees was affected by a weak external field during their development.

#### 4.6.3. Discussion

The morphological development of E-SOFI trees was demonstrated to be sensitive to a weak magnetic field. Magnetic chrome beads tended to be further towards the end of trees (by about one bead position) in the presence of a weak magnetic field than in its absence. While the force on a chrome bead was not strong enough to move the bead, the weak force likely counter-balanced the electrical forces driving the beads to the grounding electrode during the development of trees. This is a simple demonstration of the intrinsic context-sensitivity of dissipative structures, that even weak exogenous influences present in the development of dissipative structures (i.e., near the critical points) can drive substantive changes in structure. Future work will investigate whether such morphological differences have functional consequences, which would further the comparison with biology.

A key aspect of biological evolution is the intrinsic variability of heritable genetic components, in concert with the environmental activation and selection of such genes. We have argued that these properties are rudimentarily present in dissipative structures, constituted in fluctuation-amplification theory. The microscopic fluctuations of dissipative structures, like in genetics, are sensitive to external factors as well, resulting in developmental and inter-generational variability. We demonstrated such developmental variability in an electrical dissipative structure where a weak magnetic field drove morphological variants.

This discussion motivates a novel notion of selective fitness owing to (1) self-stabilizing autocatalytic dynamics and (2) developmental context sensitivity. Fitness in evolution has always depended on the survival and procreation of organisms [50,51,52]. Like organisms, dissipative structures can exhibit mutations (fluctuations) that vary phenotype (dynamical modes) which can either persist or die out. Autocatalytic processes can be differentially self-stabilizing, and thus will differentially survive, leading to rudimentary fitness. Environmental constraints can drive selection based on phenotype (phylogenetic selection) but can also drive developmental changes (ontogenetic selection) that adapt the organism to its environment. The intrinsic context-sensitivity of dissipative structures drives adaptive development. If these contextual changes enhance the system’s stability, they can be understood as conferring a selective advantage. Both the autocatalytic dynamics and intrinsic context sensitivity support notions of evolutionary fitness in non-living dissipative structures.

### 4.7. Energy Flow in Organisms and Natural Selection

The conclusion that organisms are dissipative structures, rather than machines, fundamentally changes the theoretical relationship between energy consumption (or entropy production) and biological function, with important implications for our understanding of evolution. When we consider organisms as machines, it naturally follows that minimizing irreversible processes will increase biological function (e.g., the ability of the organism to maintain itself, use energy resources effectively) and ultimately survival. Imagine two competing species, one of which uses the vast majority of its energetic resources to support motility, growth, reproduction, etc., and a second species that loses much of its available energy to heat production. Clearly, the first species will enjoy a considerable advantage and be more likely to survive, all other things being equal.

When we consider organisms as dissipative structures, it naturally follows that high levels of irreversible processes (i.e., the degradation of energy or the production of entropy) will increase the ability of organisms to maintain themselves and ultimately survive. Again, consider two competing species, one of which is superior at accessing and consuming energetic resources. Clearly, the species that consumes more resources will be more likely to survive, all other things being equal.

It is worth noting that both the conclusions above seem eminently reasonable: organisms that use energy more efficiently will enjoy an evolutionary advantage; organisms that access and consume energy at faster rates will enjoy an evolutionary advantage. However, from a physical perspective, if the processes that support biology are driven towards being optimal machines (i.e., produce entropy at minimal rates), then something additional must be at work, because machines themselves are not intrinsically driven to minimize entropy production. In fact, with repeated use, machines will become less optimal as components break down and byproducts build up. Thus, the organism-as-machine perspective must have a means to explain how and why processes are pushed away from entropy production (i.e., irreversible processes). Clearly, one could invoke evolution as the explanation here. Of course, this places another burden on evolutionary theory, as it must now explain the relationship between biological structure and energy consumption. That is, how does an organism continually organize itself such that it minimizes entropy production? Thus, the organism-as-machine perspective adds to evolutionary theory’s explanatory burden.

If we consider organisms as dissipative structures, evolutionary theory does not have to explain the relationship between structure and energy consumption, because that relationship is entailed in dissipative structure theory. That is to say, dissipative structures come into existence in the service of degrading energy potentials. They do not require any special processes to explain their origins or continued existence. By the same token, dissipative structures are in a continuous relationship with the free energy sources that they degrade. If a change in the environment or in the internal state of the structure impacts the rate at which it dissipates energy, compensatory adjustments in structure will occur, such that dissipation again returns to high levels. For example, in the E-SOFI system described above, if a tree structure is hand-set such that it has many small branches, rather than one long trunk, it will spontaneously elongate into a less branched structure (thus increasing the current). Likewise, if the flow of current to the tree structure is partially blocked, the tree will move to a location that allows for higher current. The point here is that the physics of dissipative structures is such that they intrinsically entail a self-maintaining relationship between the free energy they degrade and their own morphology and behavior. We suggest that evolution has capitalized on the remarkable properties of dissipative structures to create living systems.

## 5. Conclusions

We have aimed to provide theoretical and empirical support for the notion that the tools of nonequilibrium thermodynamics and the theory of dissipative structures can inform the theory of biological evolution. While a machine metaphor has had much pragmatic value for describing the properties of organisms, dissipative structure theory will be better suited to explain many functions of living systems. Dissipative structure theory offers a focus on process rather than state, can explain intrinsic functions (i.e., end-directedness) in nonequilibrium systems, and explains the context-sensitivity of self-organizing systems. The E- and C-SOFI systems instantiate these three aspects and demonstrate ostensibly life-like behaviors; they are structurally and functionally dynamic, they are end-directed to states of high dissipation, and they exhibit complex reorganization with small environmental changes.

The mechanisms of nonequilibrium self-organization, namely fluctuation amplification and autocatalysis, can readily be mapped on to aspects of biological evolution. The complex relationship between micro- and macroscopic variability is evident in fluctuation-amplification analysis of dissipative structures, as in the interaction between genotype and phenotype. Exogenous fluctuations can also bias the development of dissipative structures, rendering them intrinsically context-dependent. The self-replication of chemical species is apparent in abiotic complex autocatalytic chemical reactions as in the mechanisms of genetic replication. These microscopic autocatalytic processes mirror the macroscopic reproductive processes of organisms broadly. Though questions remain, we propose that treating organisms as dissipative structures, and evolution as a complex nonequilibrium process, will expand our explanatory power beyond a machine-metaphor.

## Figures and Tables

**Figure 1 entropy-22-01305-f001:**
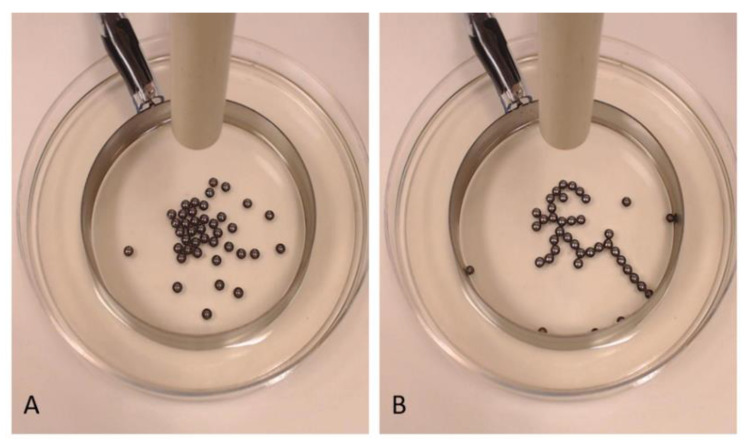
Sample images of an E-SOFI tree (**A**) Initial randomly positioned beads before formation of the tree and (**B**) after formation of the dynamic tree. The tree moves through the system like a worm foraging for higher current.

**Figure 2 entropy-22-01305-f002:**
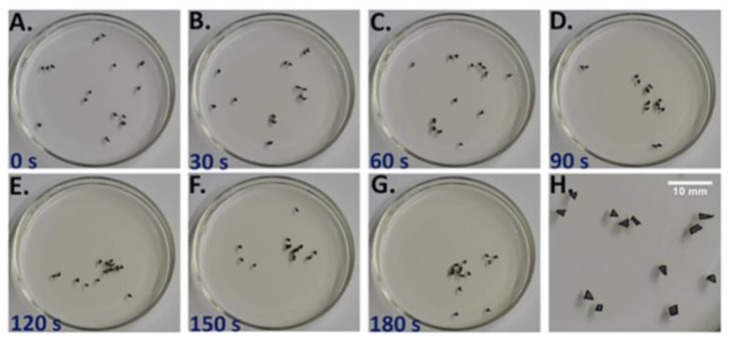
Benzoqunione pellets float at the air-water interface. Over time the pellets tend to converge near one another, creating the dynamic flock. Sub-figures show time-samples of the flock at (**A**) 0 seconds elapsed (**B**) 30 seconds elapsed (**C**) 60 seconds elapsed (**D**) 90 seconds elapsed (**E**) 120 seconds elapsed (**F**) 150 seconds elapsed (**G**) 180 seconds elapsed and (**H**) a close-up of the flock. Reprinted with permission from Satterwhite-Warden J. E., Kondepudi D. K., Dixon J. A., Rusling J. F., (2019) Thermal- and magnetic-sensitive particle flocking motion at the air-water interface. The Journal of Physical Chemistry B, 123, 3832–3840. Copyright 2019, American Chemical Society.

**Figure 3 entropy-22-01305-f003:**
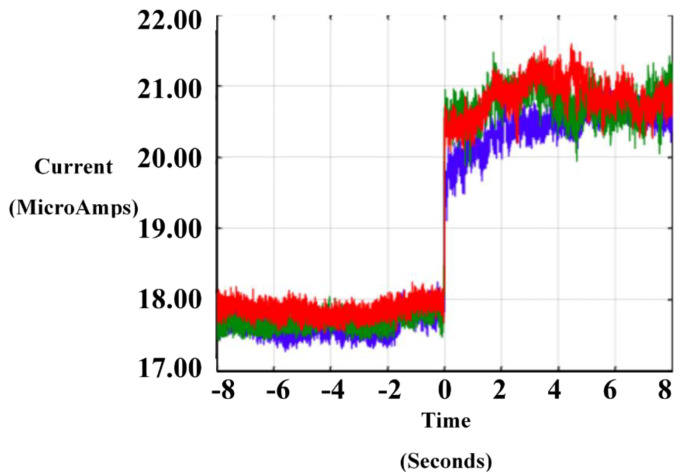
Sample profiles of the current during tree formation in the E-SOFI. Times are aligned relative to the time of formation of tree. Trees approach the approximately the same conductivity despite varying morphologies.

**Figure 4 entropy-22-01305-f004:**
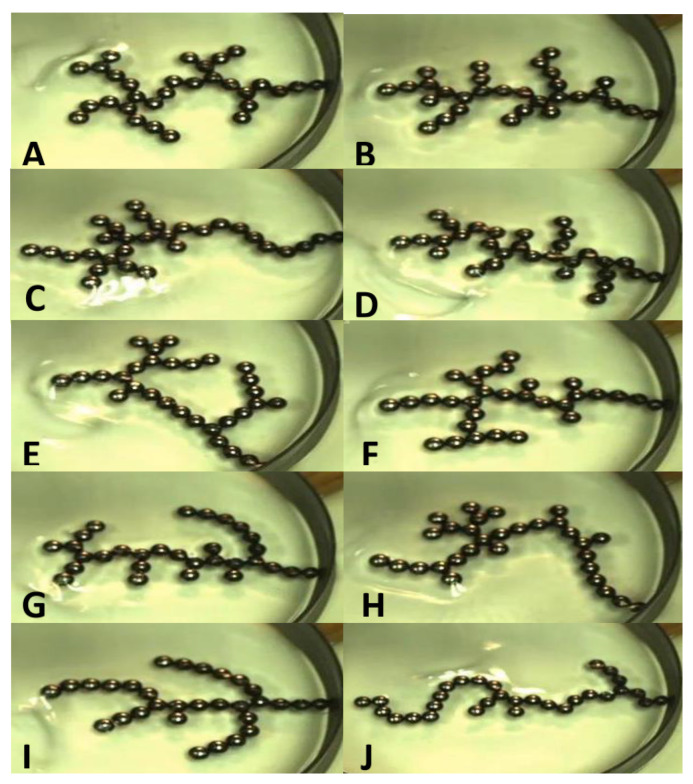
A variety of morphologies of E-SOFI trees. Each sub-figure (**A**–**J**) pictures a unique tree from a separate run of the system. Though the tree shapes are very different, they draw the same current. It is an example of variable structure resulting in the same current and hence entropy production as shown in Figure 3. Such a sample is far from exhaustive of the possibilities.

**Figure 5 entropy-22-01305-f005:**
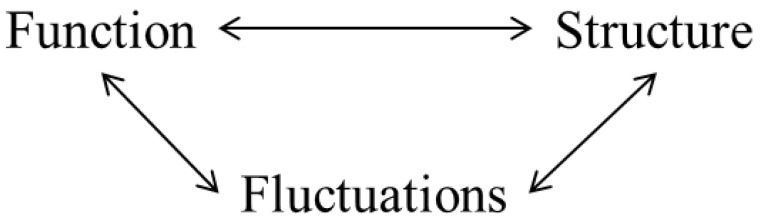
Such relations hold for both organic and inorganic dissipative structures.

**Figure 6 entropy-22-01305-f006:**
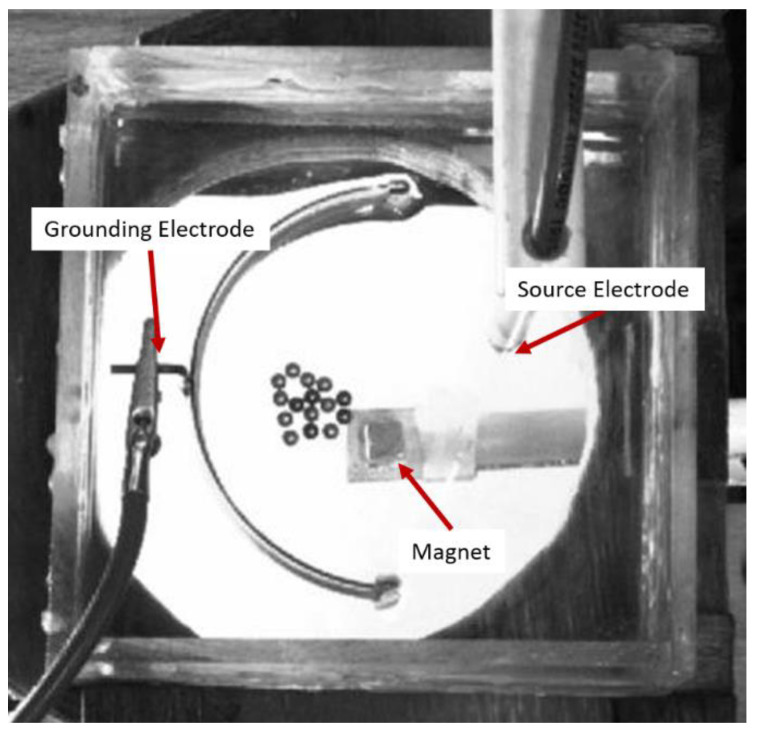
The E-SOFI setup for morphological variants. A magnet is visible below the dish. The five chrome beads are visibly darker.

**Table 1 entropy-22-01305-t001:** Machines, Dissipative Structures and Organisms.

Machines	Dissipative Structures & Organisms
Mechanics is the foundational science	Thermodynamics is the foundational science.
Structure originates from an external source by design.	Structure originates from and maintained by irreversible processes within the system.
Irreversible processes decrease functional efficiency of machines. Reducing entropy generation generally increases the efficiency of machines.	Irreversible processes are essential, without them the they would not exist. Entropy generation is an integral part of an organism’s existence. May have end directed-evolution towards state of higher entropy production.
Machines are “fractionable”	Dissipative structures are not “fractionable”.
Analysis can lead to synthesis through assembly of parts.	Analysis as sum of parts does not lead to a possibility of synthesis. Self-assembly/self-replication, not technology, is the route to synthesis.

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
