# Peer review of "Dissipative Structures, Organisms and Evolution"

_entropy, 2020, doi:10.3390/e22111305_

Round 1

Reviewer 1 Report

In this manuscript, Kondepudi and coauthors present some interesting experimental results, in particular from their E-SOFI setup, and discuss the results in terms of self-organization, dissipative structures and analogs of biology. This paper is interesting and certainly worth publishing. I have a few minor comments listed below.

Probably my major comment concerns the « machine » versus « organism » discussion. I basically agree with the authors, but I would also add that « machines » can also be considered as a part of an « organism » : our technological society. They are efficient from our anthropocentric point of view but, in helping human beings to consume more and more ressources, they obviously contribute to enhance significantly the entropy production of humankind. As underlined by the authors, « machines » are not self-replicating : they are built and « designed » for some purpose. In this sense, they are not « autonomous » and should therefore be considered as part of a larger system (human society) that may be compared to an organism. In this sense (« machines » as a small part of a bigger system), machines also contribute to maximise dissipation… and are designed to do so efficiently.

Overall, this paper presents rather classical ideas on self-organized structures, and it is not easy to identify what are the new points developped here that are really new with respect to the existing litterature. It would help to clarify to what extent the « foraging behavior » found in the E-SOFI and C-SOFI experiments is really a new feature, or if such a behavior has already been suggested as an interpretation of other experiments, … or if the authors could interpret other experiments in such terms. In any case, some background on this particular point would be useful for the reader.

Equation (1) : I do not understand the « dt » at the denominator… change in entropy is dS. If you introduce dS/dt, then this becomes a « rate of change », something which comes later.

Line 596 : « figure 4 » should be replaced by « figure 6 » ?

Reviewer 2 Report

This article is a review of the authors own experiments are theoretical work over more than 30 years, dating back to Prigogine, with emphasis on recent work.  I have several major comments. 

  1. I question the choice of the authors, to not also review similar and related work in the literature. This would not only explain and make their work stand out better, but also make the article more interesting.    For instance, the explanation of the formation of the E-SOFI trees in Figures 3 and 4 is, as far as I see it, for instance equivalent to that given by Bejan in his constructal theory (structure forms by the path of least resistance).  
  2. The authors argue that the description of dissipative structures is central to understand living systems and evolution. That this is so is evidently clear. But I fail to see that they need to introduce any new concept, that do not apply also to "machines" in Table 1. A process is also central to a chemical reactor. Take as an example  the internal structure of a fluidized bed reactor. It is also maintained via the boundaries.  Such structures may also form as a response to a changing environment, like exemplified by Fig.4.

  1. Equation 1 in the Introduction is wrong. I suppose it should be the thermodynamic definition of the entropy? Is the entropy density in Eq. 5 defined as s = S/V?  But this symbol is not used in Eqs. 6 and 7? Why are the forces and fluxes presented as J and X, when they are not used. Since the review concerns optimal states, one could wish to see the optimization problem formulated.

  1. Why are new(?) experiments reported in a review paper? They are here not well integrated in the text.

In other words, I do not see that the work brings forward much new insight. When in addition, the review is rather limited (point 1) and confusing, I conclude that the paper should not be published.

Reviewer 3 Report

Very interesting work.Self-organization far from equilibrium is very interesting topic. To understand biological organisms and their evolution, in electrically and chemically driven systems, authors showed that some dissipative structures exhibit organism-like behavior, reinforcing the earlier expectation that study of dissipative structures will provide insights into the nature of organisms and their origin. 

Author Response

We thank the reviewer for taking the time to review the manuscript.

Reviewer 4 Report

In this paper the authors discuss some of the properties of dissipative structures, give a review of their own work on “bio-analogue dissipative structures”, and discuss how the thermodynamics of these systems can provide insight into the physics of organisms and evolution. I found this paper thought provoking. Yet, there are a few points that appear overstated and other points that require refinement/clarification before publication is warranted.

1) The authors provide an unbiased discussion of MEPP in the introduction and when citing others who propose it as a driving force of evolution. However, throughout the text they assert that their E-SOFI and C-SOFI structures maximize entropy production (egs. lines 251, 296, 311-315, 336-338, and 699). It appears that none of the cited works themselves claim maximization of entropy production. In fact, the conclusions of ref [25] and [27] explicitly state that they either make no claim towards MEPP, or that they cannot experimentally confirm MEPP with their system. The authors should use similar wording to their previous papers, that their system “seeks states of higher entropy production.”

a) In section 4.6 the authors state that a dissipative interpretation of organisms would mean they respond to external changes by rebounding and maximizing dissipation (line 680). This is again assuming an MEPP interpretation. Earlier in this section, the authors ask if a mechanical interpretation of organisms is true, then “how does an organism continually organize itself such that it minimizes entropy production?” (lines 671-672). Conversely, if MEPP is true then this question could then easily become: how do they organize to maximize entropy production? Either extremum would elicit this type of question in my mind. I think the authors have some interesting points here, but the distinction is a bit unclear and perhaps muddied by the MEPP assumption.

b) Throughout the text the authors periodically discuss what would happen if the beads in the E-SOFI system are hand-set before turning on the voltage. Starting at line 680, the authors claim that pre-set small trees will evolve into larger trees that increase REP in a discussion about maximizing dissipation. If the authors want to make claims about MEPP they should perform experiments testing MEPP explicitly, for example by trying to pre-set trees with higher REP (say trees with 2-bead-wide trunks, or branches that connect – structures that might have lower resistance than the observed trees). Can such structures with lower resistance be conceived? If so, are these structures stable when the current is turned on? If these higher REP structures can be made by hand, but are not stable or not observed, this might counter an MEPP interpretation of their work.

2) Starting at line 273 the authors claim that dissipated structures have been studied for decades, “[b]ut they did not have any bio-analogues that behaved like living systems and exhibited end-directed behavior such as "foraging" and moving towards the source of energy that sustains it.” The authors are correct that there is a long history studying dissipative structures, but many systems have shown life-like behavior and end-directedness, and even behaviors like chemotaxis that also resembles foraging. Below are a few reviews that discuss some of these dissipative systems along with a few articles that explicitly highlight the life-like behaviors of their own particular systems. The most recent review article also discusses some works that resemble self-replication, which might also be applicable to the author’s comment at line 432. The authors should cite some other bio-analogues either to contrast their work or to include within their discussion.

[1] Fialkowski, M. et al. Principles and Implementations of Dissipative (Dynamic) Self-Assembly. J. Phys. Chem. B 110, 2482–2496 (2006).

[2] A. Grzybowski, B., Fitzner, K., Paczesny, J. & Granick, S. From dynamic self-assembly to networked chemical systems. Chemical Society Reviews 46, 5647–5678 (2017).

[3] Palacci, J., Sacanna, S., Steinberg, A. P., Pine, D. J. & Chaikin, P. M. Living Crystals of Light-Activated Colloidal Surfers. Science 339, 936–940 (2013).

[4] J. Solis, K. & E. Martin, J. Complex magnetic fields breathe life into fluids. Soft Matter 10, 9136–9142 (2014).

[5] Bachelard, N. et al. Emergence of an enslaved phononic bandgap in a non-equilibrium pseudo-crystal. Nature Materials 16, 808–813 (2017).

3) The discussion in sections 1.3.1 – 1.3.4 describing the characteristics of dissipative structures (amplification of fluctuations, symmetry breaking, sensing, and self-healing) should be elaborated upon as to better compare with the characteristics of evolution (1. amplification of fluctuations, 2. cooperative sensitivity, and 3. self-replication-autocatalysis). For example, it would help if terminology like cooperative/context sensitivity and autocatalysis were introduced and discussed in section 1.3.

4)  There are also a few places in the text where the discussion should be clarified.

a) The authors use the word “evolution” for two different meanings in the text. They do a good job distinguishing between the “time evolution” of dissipative systems and the “biological evolution” organisms, but there are places where this distinction is dropped. They should keep this distinction explicit throughout.

b) In the equations (1-5) the definitions of S and/or Q seem to sometimes imply the rate of change with time and other instances it is explicit with a dt in the equations. The authors should make sure every definition is consistent.

c) Table 1 is unnecessarily repetitive between the dissipative structures column and the organisms column. The authors are using it to show that organisms are complex dissipative structures, but copying and pasting table entries doesn’t add any new information. The authors might want to combine it into one “dissipative structures and organisms” column or use the organisms column to give examples.

5) Finally, I’ve noticed a few minor errors:

a) Starting at line 127, “Theory of dissipative structures…evolution”. This sentence is a main claim in the paper but it is not substantiated by the discussion in the surrounding paragraph and should be removed or moved elsewhere. It doesn’t fit here.

b) Starting at line 132: “When this happens…, a dissipative structure.” Not all far-from equilibrium systems become self-organized after becoming unstable. Say it “can” make a transition to an organized state.

c) Typo- line 94: extremum principle(s)

d) Typo- line 194: physics of organism(s)

e) Typo- line 228: are influence(d)

f) Typo- line 237: Fractionability is (an) important property

g) Typo- line 264, characteristics not referred to as A-D previously

h) Incomplete sentence- line 402: “the other pellets”… are what?

i) Typo- line 462: remove “is"

There are probably more small typos that will be found with careful rereading.

Round 2

Reviewer 2 Report

My reason to recommend rejection is based on two arguments, one formal and one scientific one.

  1. Formally speaking, I cannot understand why a paper presented as a review should report on new experimental findings. I think the authors have to choose: Either they report new original findings, in a research paper, or they report a review. If they choose for a review, I do not see why they should not review also the work of others. It is not enough to add 3 references. Presently, the review concerns their own work over the years. So, as a review, I thin the work could be better.  In their rebuttal they say they have revised the introduction to the (experimental) section to make the text run more smoothly. This does not address the real problem. 

Therefore, from this formal point of view, I recommend to reject.  

  1. Scientifically, I find the separation of a mechanical view of machines and a process-oriented view of biological systems constructed. Biological systems as well as process units in chemical or mechanical industries, can equally well be examined using the same set of differential equations (the conservation laws, the entropy balance and the constitutive equations).  Still, we know and appreciate that biological systems are more complicated and prone to other control mechanisms than non-living systems. Others like for instance Bejan, have shown that “evolution-like phenomena can be instantiated in non-living dissipative structures”.  

When the authors claim that “ We present the profound difference between machines, based mechanics on the one hand, and dissipative structures and organisms founded on thermodynamics on the other.  Even today life is thought of as simply a complex machine and an organism thought of as an assembly of parts" , I like to dispute these statements.

I do not dispute their experimental results, but the conclusion drawn from them.  In my mind, the authors have to argue much better to document the profound difference.   

Therefore, from this scientific point of view, I also recommend to reject the paper.
